# Atherogenic Index of Plasma in Obstructive Sleep Apnoea

**DOI:** 10.3390/jcm10030417

**Published:** 2021-01-22

**Authors:** Andras Bikov, Martina Meszaros, Laszlo Kunos, Alina Gabriela Negru, Stefan Marian Frent, Stefan Mihaicuta

**Affiliations:** 1Department of Pulmonology, Semmelweis University, 1085 Budapest, Hungary; andras.bikov@gmail.com (A.B.); martina.meszaros1015@gmail.com (M.M.); laszlo.kunos@gmail.com (L.K.); 2North West Lung Centre, Wythenshawe Hospital, Manchester University NHS Foundation Trust, Manchester M23 9LT, UK; 3Division of Infection, Immunity & Respiratory Medicine, University of Manchester, Manchester M23 9LT, UK; 4Department of Cardiology, University of Medicine and Pharmacy Timisoara, 300041 Timisoara, Romania; eivanica@yahoo.com; 5Cardiology Department, Institute of Cardiovascular Diseases Timisoara, 300310 Timisoara, Romania; 6Department of Pulmonology, University of Medicine and Pharmacy Timisoara, 300041 Timisoara, Romania; stefan.mihaicuta@umft.ro; 7Pulmonology Department, ‘Victor Babes’ Infectious Diseases and Pulmonology Hospital Timisoara, 300310 Timisoara, Romania

**Keywords:** sleep apnea, atherogenic index of plasma, dyslipidemia, cardiovascular disease

## Abstract

Background: Dyslipidaemia is well recognised in obstructive sleep apnoea (OSA) and could contribute to the development of cardiovascular disease (CVD). Atherogenic index of plasma (AIP) predicts cardiovascular morbidity and mortality better than the individual lipid levels. The aim of this study was to investigate the AIP in patients with OSA in relation with disease severity. Methods: Four hundred sixty-one patients with OSA and 99 controls participated in this study. AIP was assessed in the morning following a diagnostic sleep study. The association between lipid values and OSA were adjusted for age, gender, and body mass index. Results: Patients with OSA had higher AIP and triglyceride, and lower high-density lipoprotein cholesterol (HDL-C) levels (all *p* < 0.05). AIP significantly correlated with the Epworth Sleepiness Scale score (*ρ* = 0.19), the apnoea-hypopnoea index (*ρ* = 0.40) and oxygen desaturation index (*ρ* = 0.43, all *p* < 0.05). However, there was no relationship between the AIP and markers of sleep quality such as total sleep time, sleep period time, sleep efficiency, arousal index or percentage of REM sleep (all *p* > 0.05). AIP was not a better predictor for self-reported cardiovascular disease or diabetes than HDL-C. Conclusions: AIP is elevated in OSA and is related to disease severity. However, it does not seem to have an additional clinical value compared to HDL-C.

## 1. Introduction

Obstructive sleep apnoea (OSA) is a common disease which is characterised by a repetitive collapse of the upper airways during sleep resulting in chronic intermittent hypoxaemia and frequent microarousals. OSA is a well-known risk factor for cardiovascular and cerebrovascular disease [1]; however, the treatment of OSA with continuous positive airway pressure (CPAP) has only a marginal effect on cardiovascular mortality [2]. Therefore, understanding mechanisms leading to cardiovascular disease and identifying clinical markers of disease development are of clinical importance.

Dyslipidaemia is an essential component in the pathophysiology of atherosclerosis [3]. Low-density lipoprotein (LDL) particles directly responsible for atherosclerosis [3,4]. However, high triglyceride (TG) levels are also independently related to cardiovascular risk [5]. Although high-density lipoprotein cholesterol (HDL-C) levels may not directly be related to cardiovascular risk according to the mendelian randomisation studies, HDL-C values lower than 2.3 mmol/L increase the accuracy of the cardiovascular risk evaluation. Moreover, HDL-C levels are suggested to be taken into consideration in Systematic Coronary Risk Evaluation (SCORE) model [6].

Obstructive sleep apnoea is associated with dyslipidaemia [7,8]. More importantly, OSA can further accelerate dyslipidaemia-related atherosclerosis by the oxidation of LDL particles [3]. Several mechanisms are responsible for the development of dyslipidemia in OSA [7]. First, patients with OSA tend to consume high-calorie and high-fat food diet [9]. Second, postprandial TG clearance is impaired due to reduced lipoprotein lipase activity [10]. Third, hepatic TG production is increased due to chronic intermittent hypoxaemia [11]. Fourth, the liberation of free fatty acids from adipose tissue is accelerated by the sympathetic bursts associated with OSA [12]. Finally, the reverse cholesterol transport is impaired in OSA due to reduced production of apolipoprotein A [13]. These studies indicate that rather than focusing on a single component of dyslipidaemia, comprehensive biomarkers are warranted for the assessment of lipid abnormalities and subsequent cardiovascular risk in OSA.

The logarithmically transformed ratio of TGs to HDL-C, known as the atherogenic index of plasma (AIP), has recently come to attention as a biomarker of cardiovascular disease [14,15]. One merit of AIP is that it also reflects on the esterification rate of HDL particles and therefore allows for a more accurate assessment of cardiovascular risk than the HDL-C measurement itself [16]. Indeed, AIP predicts cardiovascular morbidity [14] and mortality [17] better than the single-lipid components. However, although AIP is a marker for cardiovascular morbidity and mortality, the complex interplay between OSA, which is also a demonstrated risk factor, dyslipidaemia and cardiovascular disease, is not yet fully understood.

Only six studies have investigated TG/HDL-C ratio or AIP in OSA [18,19,20,21,22,23], but only four of them compared patients with OSA to non-OSA controls [14,15,21,23]. Three of these studies were conducted in Eastern Asia [14,15,21] and one in Brazil [23]. As the relationship between OSA and dyslipidaemia is influenced by genetic factors [24], lifestyle, diet, medications and comorbidities [7,8,25], it is reasonable to assume that the conclusions of the previous studies cannot be fully extrapolated to a Central European cohort. Furthermore, no studies evaluated AIP in OSA in Central Europe.

Therefore, the primary objective of the present study was to assess if there are any differences in AIP between patients with OSA and controls and if there is a correlation with disease severity. Our exploratory aims included the assessment of the relationship between AIP and parameters of sleep quality, the effect of statin therapy on the relationship between AIP and OSA and to compare the predictive utility of AIP with other lipid parameters.

## 2. Methods

### 2.1. Study Subjects and Design

Five hundred and sixty consecutive volunteers participated in this dual-centre study. They were originally referred for a diagnostic sleep study due to suspected obstructive sleep apnoea (i.e., snoring, pauses in the breathing, and daytime tiredness). None of them had received any treatment for OSA, such as CPAP or mandibular advancement device. Patients with acute respiratory, heart or renal failure were excluded. OSA was diagnosed in 461 patients (apnoea-hypopnoea index, AHI ≥ 5/h) based on inpatient cardiorespiratory polygraphy (PG, *n* = 102) or polysomnography (PSG, *n* = 458). We used the International Classification of Sleep Disorders (Third Edition) criteria for the diagnosis of sleep apnoea (i.e., AHI ≥ 5/h with day- or night-time symptoms or associated with comorbidities). Subjects in whom OSA was excluded (AHI < 5/h) comprised the control group (*n* = 99).

Following informed consent, patients filled out the Epworth Sleepiness Scale (ESS) questionnaire and a detailed medical history was taken, which included smoking habits and comorbidities. Comorbidities were defined based on patients’ report, prescriptions, and available medical records. In the event of diagnosis uncertainty, patients were referred for further diagnostic testing.

Fasting venous blood samples were taken in the morning following the diagnostic sleep test in order to assess the serum levels of TG, total cholesterol (TC), LDL-C and HDL-C. AIP was calculated as log_10_ (TG/HDL-C), with TGs and HDL-C both expressed in mmol/L.

The study was approved by the local Ethics Committee (Semmelweis University TUKEB 30/2014 and RKEB 172/2018, and 22/2014/24.07.2019 University of Medicine and Pharmacy Victor Babes Timisoara) and patients gave their informed consent before participating in the study.

### 2.2. Sleep Studies

Inpatient cardiorespiratory PG and PSG were performed according to the American Academy of Sleep Medicine (AASM) recommendations [26]. Sleep stages, movements and cardiopulmonary events were scored manually according to AASM guideline [27]. Apnoea was defined as ≥90% drop in the nasal flow lasting for ≥10 s. Hypopnoea was defined as ≥30% drop in the nasal flow lasting for ≥10 s which is associated with either ≥3% drop in the oxygen saturation (for both PG and PSG) or arousal (for PSG). We recorded total sleep time (TST), sleep period time (SPT), percentage of rapid eye movement sleep (REM%) and minimal oxygen saturation (MinSatO2) and calculated Sleep efficiency (Sleep%, TST/SPT), AHI, oxygen desaturation index (ODI), arousal index (AI) and the percentage of total sleep time spent with saturation below 90% (TST90%).

### 2.3. Statistical Analyses

JASP 0.14 (JASP Team, University of Amsterdam, Amsterdam, The Netherlands) and MedCalc 19.5.3 (MedCalc Software Ltd., Ostend, Belgium) were used for statistical analysis. The normality of the data was assessed with the Shapiro-Wilk test. The OSA and control groups were compared with Mann-Whitney and Chi-square tests. The relationship between TGs, TC, LDL-C, HDL-C and AIP as well as OSA was investigated with multivariate logistic regression analysis adjusted for age, gender, and body mass index (BMI). The relationship between lipids and OSA severity was investigated with non-parametric analysis of covariance (ANCOVA) test adjusted for age, gender and BMI. The relationships between lipids and sleep parameters were studied with Spearman’s test. The predictive value of AIP for OSA, cardiovascular disease, hypertension and diabetes was evaluated with the Receiver operating characteristic (ROC) analysis and compared to TGs, LDL-C and HDL-C with the DeLong test [28]. A sensitivity analysis on the primary outcomes was performed in people not taking any statin (*n* = 348 OSA and 94 controls) and in those who had PSG as a diagnostic test. Data are expressed as median /interquartile range/. A *p* value < 0.05 was considered significant.

## 3. Results

### 3.1. Comparison of the OSA and Control Groups

As expected, patients with OSA were older, had higher BMI, had a higher percentage of males, smokers, and comorbidities, except for diabetes and COPD which did not reach the level of significance. Patients with OSA more likely took statins, had higher TG, AIP and lower HDL-C levels. In addition, patients with OSA had higher ESS, longer TST and SPT, AHI, ODI and TST 90% values, and had lower MinSatO2 as well as poorer sleep efficiency. There was no difference in TC, LDL-C or REM% between the two groups. The results are summarised in Table 1.

### 3.2. The Relationship between Lipids and OSA

After adjusting for age, gender and BMI, higher AIP, lower HDL-C, as well as, surprisingly, lower TC levels were associated with OSA. There was no relationship between OSA and TG or LDL-C levels (Table 2).

One hundred sixteen patients were diagnosed with mild (AHI 5–14.9/h), 125 with moderate (AHI 15–29.9/h) and 220 with severe (AHI ≥ 30/h) disease. Investigating lipid levels according to OSA severity, there was no difference in TC (*p* = 0.21) or LDL-C (*p* = 0.77), while progressively increasing triglyceride (*p* < 0.001) and AIP (*p* < 0.001) and progressively decreasing HDL-C (*p* < 0.001) levels were noted in OSA (Figure 1).

There was a significant correlation between HDL-C levels and ESS (*ρ* = −0.26), SPT (*ρ* = −0.131), AHI (*ρ* = −0.40), ODI (*ρ* = −0.43), TST90% (*ρ* = 0.32) and MinSatO2 (*ρ* = 0.27); between TG levels and ESS (*ρ* = 0.15), AHI (*ρ* = 0.32), ODI (*ρ* = 0.43), TST90% (*ρ* = 0.31) and MinSatO2 (*ρ* = −0.25); as well as between AIP and ESS (*ρ* = 0.19); AHI (*ρ* = 0.40), ODI (*ρ* = 0.43), TST90 (*ρ* = 0.36) and MinSatO2 (*ρ* = −0.28, all *p* < 0.05). TC or LDL-C did not correlate with any of the sleep parameters. Parameters of sleep quality did not correlate with lipid values, except for the indirect association between HDL-C and SPT as mentioned above. Interestingly, we also found a significant direct correlation between the arousal index (AI) and HDL-C in OSA group (*ρ* = 0.112, *p* = 0.028). No correlation was found between AI and any other lipid values, including AIP.

Checking the performance of lipid parameters to distinguish patients with OSA from controls HDL-C, TGs and AIP were found to be significant predictors based on ROC analysis (all *p* < 0.05). The areas under the curves (AUC) with their 95% confidence intervals were 0.816/0.769–0.863/for HDL-C, 0.707/0.649–0.765/for TGs and 0.778/0.723–0.833/for AIP. Comparing the areas under the curves, AUC of HDL-C was significantly greater than of AIP (*p* = 0.03) and of TGs (*p* < 0.001) and the AUC of AIP was significantly greater than of TGs (*p* < 0.001, Figure 2).

### 3.3. The Relationship between Lipids and OSA in Subjects Not Taking Statins

In subjects not taking statins, following adjustment for age, gender and BMI, higher AIP and lower HDL-C levels were associated with OSA. There was no relationship between OSA and TC, TGs or LDL-C (Table 3).

Ninety-three controls, 88 patients with mild OSA, 91 patients with moderate OSA and 169 patients with severe disease did not take statins. Similarly to the whole study population, increasing disease severity was associated with progressively increasing TGs (*p* < 0.001) and AIP (*p* < 0.001), and progressively decreasing HDL-C (*p* < 0.001) in this subgroup. There was no correlation between OSA severity and TC (*p* = 0.28) or LDL-C (*p* = 0.78) levels.

Similarly to the entire study population, AUC of HDL-C (0.815/0.776–0.850/), TGs (0.704/0.659–0.747/) and AIP (0.771/0.729–0.810/) were significant predictors for OSA (all *p* < 0.05). Comparing the AUCs, the AUC of HDL-C was greater than that of AIP (*p* = 0.01), and both of them were significantly greater than the AUC of TGs (both *p* < 0.001).

### 3.4. The Relationship between Lipids and OSA in Subjects Who Had PSG as a Diagnostic Test

Higher AIP, as well as lower HDL-C and TC levels were associated with the presence of OSA in this subgroup as well. There was no relationship between OSA diagnosis and TG or LDL-C levels (Table 4).

Similarly to the whole cohort, increasing OSA severity was associated with increasing TG and AIP and decreasing HDL-C levels (all *p* < 0.001). There was no correlation between OSA severity and TC (*p* = 0.21) or LDL-C (*p* = 0.55).

In subjects who had PSG as a diagnostic test, HDL-C (0.824/0.786–0.858/), TG (0.696/0.651–0.738/) and AIP (0.763/0.721–0.801) were significant predictors for OSA (all *p* < 0.05). Similarly, to the analysis in all subjects, the performance of HDL-C to distinguish patients with OSA from controls was superior to AIP and TGs (both *p* < 0.01), while the AUC of AIP was greater than of TGs (*p* < 0.001).

### 3.5. Atherogenic Index of Plasma as a Predictor for Self-Reported Hypertension, Diabetes as Well as Cerebrovascular or Cardiovascular Disease in OSA

In patients with OSA, all lipids except for LDL-C (*p* = 0.08) were predictive for cardiovascular or cerebrovascular disease (all *p* < 0.05). However, there was no difference between their AUCs (all *p* values for all comparisons >0.05). HDL-C, TGs and AIP were predictive for diabetes, without any difference in their AUCs (*p* > 0.05). Evaluating hypertension, only HDL-C was predictive (*p* = 0.03, Table 5).

Statin therapy or PSG as a diagnostic test did not change the conclusions above, as AIP was not superior to predict these disorders in OSA in neither of these subgroups (*p* values for all comparisons >0.05).

## 4. Discussion

Obstructive sleep apnoea is a common disease and a significant risk factor for cardiovascular disease [29]. The burden of asymptomatic coronary artery plaques may be as high as 80% in OSA [30]. This highlights the importance of analysing biomarkers for cardiovascular disease in patients with OSA as part of the routine clinical practice. In this study, we evaluated the atherogenic index of plasma, a promising marker for cardiovascular morbidity and mortality, and demonstrated that its value is elevated in OSA and correlates with disease severity.

The TGs/HDL-C ratio or its logarithm has been investigated only in a limited number of studies in OSA. Shumizu et al. recruited 1528 Japanese male subjects, including 241 patients with OSA. They found that OSA was associated with elevated AIP but only in normal-weight subjects (BMI < 25 kg/m^2^) [18]. Wu et al. investigated 246 male bus drivers from Taiwan. TGs/HDL-C ratio was associated with OSA and correlated with disease severity [19]. The obvious criticism of these two studies is that they investigated men exclusively, while the association between OSA and dyslipidaemia, AIP in particular, depends on gender as well [20]. Cao et al. studied 284 Chinese subjects, 256 were diagnosed with OSA. They found that AIP was higher in OSA and was related to AHI [21]. Wysocki et al. investigated 211 patients with OSA and found a significant relationship between AHI and AIP; however, the authors did not recruit any controls [22]. Finally, Silva et al. compared patients with mild OSA to controls and found increased TGs/HDL-C ratio in the OSA group. TGs/HDL-C ratio did not correlate with any parameters of sleep quality and was not related to sleepiness [23]. The current study is in line with the previously published literature, reporting that AIP is elevated in OSA and correlates with disease severity. Similarly to the report of Silva et al., the lipid values did not correlate with sleep quality parameters in our study, except for the correlation between the HDL-C and SPT as well as AI. However, compared to the previous studies, our cohort had a wider representation of age, gender, comorbidities, ratio of patients and controls and, most importantly, the severity of OSA.

The two cardinal features of OSA are chronic intermittent hypoxaemia and disruption of the sleep architecture. It is likely that hypoxaemia is the predominant mechanism responsible for dyslipidaemia in OSA. However, disturbed sleep may contribute via hyperphagia as well [8]. In support of this, similarly to the findings by Silva et al. [23], we did not find any relationship between the parameters of sleep quality and AIP. However, the study by Silva et al. investigated only patients with mild OSA. We believe that our results on a population with a wider range of disease severity extend the previous results. In contrast to the study by Silva et al. [23], daytime sleepiness was related to higher AIP in our report. However, in this study half of the patients with OSA had severe disease. It is well-known that ESS score increases with disease severity [31]. Moreover, patients with excessive daytime sleepiness are also characterised by higher levels of hypoxia [32], which is related to the severity of OSA and leads to impaired lipid homeostasis. Thus, it has been postulated that excessive daytime sleepiness may independently predict cardiovascular risk [33]. It has also been reported that obstructive respiratory events occurring during REM sleep do not exacerbate more lipid abnormalities compared to the events occurring in non-REM sleep [13,34]. Our data expand this knowledge by reporting no association between AIP and REM%.

Statins are the most widely used lipid-lowering medications. Apart from lowering total cholesterol, they may elevate HDL-C concentrations; however, this effect differs between drugs [35]. Due to this potential bias, in our sensitivity analysis, we excluded subjects who were taking statins. Most of our results very compatible with those in the whole group, except that reduced total cholesterol levels were no longer significantly associated with OSA in the adjusted model. Lower TC levels in the OSA group were, therefore, likely due to an increased usage of statins in these patients. In the previous studies, patients on statins were either included [19,22], excluded [21], or the results were adjusted for statin usage [18,23]. Nevertheless, excluding patients on statins did not change our conclusions on AIP.

Although both PSG and cardiorespiratory polygraphy are validated tests to diagnose OSA, PSG is a more sensitive and specific tool, as it measures the sleep time precisely and takes into account arousals when defining hypopneas [36]. Most of the previous studies used PSG [19,21,22,23]. To make our results more comparable with the previous one, we performed a separate analysis on the data from subjects who had polysomnography as a diagnostic test. The results were in line with those obtained in the whole group of volunteers.

AIP indicates a pathological dysregulation between anti-atherogenic and atherogenic lipoproteins. AIP was described as a marker of atherogenic oxidated small dense LDL [15]. Thus, it has a predictive power to evaluate cardiovascular risk. We detected an association between AIP and the markers of overnight hypoxia. It was also suggested that AIP provides more precise information when used simultaneously with single lipid components [16]. Low HDL-C levels also emerged as a risk factor for cardiovascular consequences [6]. Surprisingly, AIP did not show a stronger association with OSA than HDL-C in our study population. Our results are in line with the previous findings which did not find a difference in the strength of association with OSA between HDL-C and AIP [19,21]. In contrast, Silva et al. found that AIP but not HDL-C was different in OSA compared to controls; however, they included only mild subjects [23]. More surprisingly, AIP did not show a better predictive value for self-reported cardiovascular disease or diabetes compared to other lipid components in OSA. Of note, these conditions were diagnosed before the study; hence, administration of specific medications and adoption of healthy lifestyle measures may explain our findings. Prospective, follow-up studies in OSA are warranted to investigate whether AIP could serve as a better predictor for the development of these disorders than the individual components.

Previous studies suggested that obesity could be a significant covariate determining the relationship between AIP and OSA [18,19]. However, a large European study found that the relationship between OSA and TGs, as well as HDL-C, was independent of obesity [37]. Our results are in line with this finding, reporting no change in this relationship after adjusting for BMI. Of note, central obesity (hip-waist ratio) may be a better predictor for metabolic outcomes than the BMI. Unfortunately, these results were not recorded in this study. Most importantly, our results were adjusted for gender and age, which could also contribute to metabolic syndrome in OSA [38].

Interestingly, OSA was not associated with total cholesterol or LDL-C levels. This is in line with the results reported from the European Sleep Apnea Database, suggesting that HDL-C and triglyceride better characterise OSA-related dyslipidaemia than other lipid parameters [37]. A potential reason for this could be lifestyle-related factors; however, we did not assess these in our current study.

Although fasting blood was taken for lipid measurements, we did not control for diet, alcohol consumption or regular exercise which could be a potential bias for our study. Although not routinely recommended to fast before lipid profile measurement, diet essentially contributes to the metabolic profile [39]. We acknowledge this as a limitation of our investigation. In our study PSG has not been performed in every participant. PSG is a more sensitive test to score hypopnoeas resulting arousals compared to PG; thus, PG may misclassify the severity of the disease. The study had a cross-sectional case-control design and further follow-up observational studies or randomised clinical trials with CPAP are warranted to better describe the clinical value of AIP in OSA. We believe our study will provide a basis to design such prospective trials. The comorbidities in our study were self-reported which limits the conclusion of the relationship between lipid parameters and their prevalence. The control group consisted of subjects having suggestive symptoms for sleep-disordered breathing, but a negative sleep study for the diagnosis of OSA. Inclusion of subjects not reporting any symptoms could have provided different results. In addition, age, gender and BMI were different between the OSA and control groups. Although the analyses were adjusted for these factors, further studies balancing the OSA and control groups on these parameters are warranted to explore the contribution of OSA on AIP. Finally, our study demonstrates that OSA is related to altered AIP; however, to assess if OSA is a confounding factor when using AIP for predicting cardiovascular morbidity and mortality in general population, follow up studies are needed.

## 5. Conclusions

In summary, we reported that AIP is increased in OSA and related to disease severity, but not the quality or quantity of sleep. Although this was only an exploratory objective, we could not prove the superiority of AIP over other lipid parameters in assessing dyslipidaemia in OSA. Nevertheless, as OSA significantly influences AIP, the role of AIP as a predictive biomarker for cardiovascular disease in the general population may be affected by the presence of obstructive sleep apnoea. This needs to be investigated in follow-up studies.

## Figures and Tables

**Figure 1 jcm-10-00417-f001:**
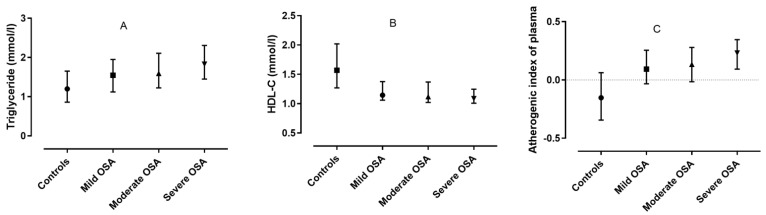
The relationship between lipid fractions and increasing OSA severity: **A**. Triglyceride; **B**. HDL-C; **C**. Atherogenic index of plasma.

**Figure 2 jcm-10-00417-f002:**
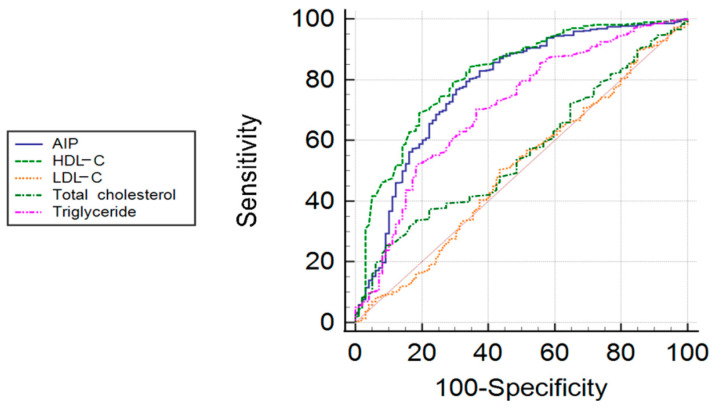
Receiver operating characteristic curve for specific lipid fractions to detect OSA.

**Table 1 jcm-10-00417-t001:** Comparison of OSA and control groups—demographic, anthropometric, clinical and sleep characteristics.

	OSA (*n* = 461)	Control (*n* = 99)	*p*-Value
Age (years)	54/46–62/	46/34.5–59.5/	<0.001
Gender (% male)	65	31	<0.001
BMI (kg/m^2^)	31.8/28.3–35.8/	24.8/21.7–28.1/	<0.001
Smokers (ever%)	42	14	<0.001
Cigarette pack years	0/0–15/	0/0–0/	<0.001
Hypertension (%)	68	38	<0.001
Cardiovascular or cerebrovascular disease (%)	29	5	<0.001
Cardiac arrhythmia (%)	26	12	0.002
Diabetes mellitus (%)	20	13	0.09
COPD (%)	12	6	0.08
Statin users (%)	24	6	<0.001
TGs (mmol/L)	1.7/1.3–2.1/	1.2/0.9–1.6/	<0.001
TC (mmol/L)	5.1/4.3–5.9/	5.2/4.7–6.1/	0.061
LDL-C (mmol/L)	2.9/2.4–3.8/	3.1/2.4–3.8/	0.868
HDL-C (mmol/L)	1.1/1.0–1.3/	1.6/1.3–2.0/	<0.001
AIP	0.18/0.03–0.31/	−0.15/−0.34–0.06/	<0.001
ESS	8/5–11/	6/4–9/	<0.001
TST (min) *	423/375–463/	400/358–425/	0.003
SPT (min) *	470/431–503/	423/397–444/	<0.001
Sleep% *	91/82–97/	95/88–99/	0.008
REM% *	15.1/11.6–20.2/	16.1/12.2–20.8/	0.946
AHI (1/h)	28.6/14.9–47.7/	2.3/1.2–3.5/	<0.001
ODI (1/h)	26.0/13.5–47.9/	1.1/0.4–2.2/	<0.001
TST90% (%)	7.3/1.3–25.4/	0/0–0.1/	<0.001
MinSatO_2_ (%)	82/75–87/	91/88–93/	<0.001

AHI—apnoea hypopnoea index; AIP—atherogenic index of plasma; BMI—body mass index; COPD—chronic obstructive pulmonary disease; ESS—Epworth Sleepiness Scale; HDL-C—high-density lipoprotein cholesterol; LDL-C—low-density lipoprotein cholesterol; MinSatO2—minimal oxygen saturation; ODI—oxygen desaturation index; REM%—the percentage of REM sleep of the total sleep time; SPT—Sleep period time; Sleep%—sleep efficiency; TC—total cholesterol; TGs—triglycerides; TST—total sleep time; TST90%—the percentage of total sleep time spent with oxygen saturation below 90%. * Estimated in 488 subjects.

**Table 2 jcm-10-00417-t002:** The relationship between lipidic fractions and OSA.

	B	*p*
TGs (mmol/L)	0.33	0.07
TC (mmol/L)	−0.27	0.02
LDL-C (mmol/L)	−0.07	0.24
HDL-C (mmol/L)	−1.02	<0.001
AIP	2.02	<0.001

AIP—atherogenic index of plasma; HDL-C—high-density lipoprotein cholesterol; LDL-C—low-density lipoprotein cholesterol; TC—total cholesterol; TGs—triglycerides.

**Table 3 jcm-10-00417-t003:** The relationship between lipidic fractions and OSA in subjects not taking statins.

	*β*	*p*-Value
TGs (mmol/L)	0.30	0.12
TC (mmol/L)	−0.23	0.054
LDL-C (mmol/L)	−0.07	0.21
HDL-C (mmol/L)	−0.84	0.02
AIP	1.81	<0.001

AIP—atherogenic index of plasma; HDL-C—high-density lipoprotein cholesterol; LDL-C—low-density lipoprotein cholesterol; TC—total cholesterol; TGs—triglycerides.

**Table 4 jcm-10-00417-t004:** The relationship between lipidic fractions and OSA in subjects undergoing diagnostic PSG.

	*B*	*p*-value
TGs (mmol/L)	0.12	0.49
TC (mmol/L)	−0.34	0.02
LDL-C (mmol/L)	−0.05	0.45
HDL-C (mmol/L)	−0.98	0.002
AIP	1.79	0.004

AIP—atherogenic index of plasma; HDL-C—high-density lipoprotein cholesterol; LDL-C—low-density lipoprotein cholesterol; TC—total cholesterol; TGs—triglycerides.

**Table 5 jcm-10-00417-t005:** The predictive value of lipidic fractions for cerebrovascular/cardiovascular disease, diabetes mellitus and arterial hypertension in patients with OSA.

	Cerebrovascular and Cardiovascular Disease			Diabetes Mellitus		Arterial Hypertension	
	AUC	95% CI	AUC		95% CI	AUC	95% CI
AIP	**0.604**	**0.558 to 0.649**	**0.627**		**0.581 to 0.671**	0.553	0.506 to 0.599
HDL-C	**0.586**	**0.539 to 0.631**	**0.579**		**0.532 to 0.625**	**0.561**	**0.515 to 0.607**
LDL-C	0.551	0.505 to 0.598	0.560		0.513 to 0.606	0.507	0.460 to 0.553
TC	**0.563**	**0.517 to 0.609**	0.522		0.475 to 0.568	0.510	0.463 to 0.556
TGs	**0.601**	**0.555 to 0.646**	**0.626**		**0.580 to 0.670**	0.536	0.489 to 0.582

AIP—atherogenic index of plasma; AUC—area under the curve; CI—confidence intervals; HDL-C—high-density lipoprotein cholesterol; LDL-C—low-density lipoprotein cholesterol; TC—total cholesterol; TGs—triglycerides. Significant AUCs are highlighted in bold.

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
