# Peer review of "Atherogenic Index of Plasma in Obstructive Sleep Apnoea"

_jcm, 2021, doi:10.3390/jcm10030417_

Round 1

Reviewer 1 Report

The authors present the results of a study that evaluates the atherogenic index of plasma (AIP) in patients with OSA and its association with OSA presence and severity. Moreover the authors also assess if AIP could predict other comorbidities. There are previous studies previous on this topic and therefore the novelty of the findings is not high and what the study adds to previous data is not clear.

I belive that there are some questions that should be adressed:

1. From the current information, it is not entirely clear the inclusion criteria and exclusion criteria, how the subjects were included (consecutively?) and how the patient flow was. A patient flow showing patient/control disposition and also a better definition of inclusion and exclusion criteria would be appreciated.

 2. How the authors define OSA should be included in the methods section as well as a reference related to the definition used. Why did the authors decide to define OSA based on AHI > 5/h in patients without significant somnolence instead of AHI>15/h?

3. As suggested in table 1, OSA patients have preserved sleep efficiency and percentage of REM sleep and hypoxemia indices are not very high, suggesting that they have not a severe disease. Moreover they do not present somnolence. It does not seem the typical profile of subjects with OSA suspicion. Please give your opinion on that point.

 4. Analysis were adjusted by gender, age and BMI, but other factors that could influence the results such as exercise, diet, dyslipidemia treatment (statin or others), comorbidities or waist and hip circumference have not been included. Please provide the analysis adjusted by these other factors. 

5.Please provide a more detailed discussion on the association between lipid parametres and OSA and cormorbidities. Why the authors believe there is association between OSA and HDL-C or TG and not with other parametres i.e. TC or LDL-C? Why do you explain that some of these parameters are correlated with the EES while other authors did not find this association? Why do you believe that HDL-C is predictive of hypertension? 

6.The main two reasons why a sleep disorder such as OSA is related to CV events are the intermitten hypoxia and sleep fragmentation. Indeed, most of the patients were diagnosed with polysomnography, there is no data on sleep fragmentation assessed by the arousal index. Please provide this information and its association with AIP.

7. Two types of sleep test were used. Therefore there is the possibility that the respiraory polygrapy could understimate or misclassify the severity of the disease. This aspect should be included in the limitation section.

8. Comorbidities are self-reported and subjects could be receiving treatment for that conditions. This is a huge limitation to conclude associction between AIP and comorbidities and it could difficult the extrapolation. Please ellaborate more on the matter.

9. In the discussion, I think the authors should go into a bit more detail into the consequences and clinical implications of the finding.

 Author Response

Reviewer 2 Report

Dear sirs,

perhaps it would have been more appropriate to collect the control group from patients without symptoms suggestive for OSA and negative for PSG.              The group of OSAS patients has a higher BMI, has more male and older subjects than the control group. It also makes more use of statins. Already from these characteristics, the result of the significance of AIP and low HDL levels in the OSAS group was predictable.                                                      Why have some patients been studied with PSG (458) and others by cardiorespiratory monitoring (102) ?   What is the reason behind this discrepancy ?                                                                                           As you rightly pointed out, there are studies that correlate AIP with cardiovascular morbidity and mortality. What happens about this relationship in the patient with OSAS ?                                                                       This will need to be evaluated in future studies to quantify the importance of AIP study in the OSAS patient.                                                                 Your study shows the correlation between AIP and OSAS severity. What are the clinical implications? It is not clear from the article
